# Built-In Packaging for Two-Terminal Devices

**DOI:** 10.3390/mi14071473

**Published:** 2023-07-22

**Authors:** Ahmet Gulsaran, Bersu Bastug Azer, Dogu Ozyigit, Resul Saritas, Samed Kocer, Eihab Abdel-Rahman, Mustafa Yavuz

**Affiliations:** 1Mechanical and Mechatronics Engineering Department, University of Waterloo, Waterloo, ON N2L 3G1, Canada; bbastuga@uwaterloo.ca (B.B.A.); dozyigit@uwaterloo.ca (D.O.); 2Waterloo Institute for Nanotechnology (WIN), University of Waterloo, Waterloo, ON N2L 3G1, Canada; resulsaritas@gmail.com (R.S.); skocer@uwaterloo.ca (S.K.); eihab@uwaterloo.ca (E.A.-R.); 3Systems Design Engineering Department, University of Waterloo, Waterloo, ON N2L 3G1, Canada

**Keywords:** packaging, wire bonding, diode application, metal/insulator/metal (MIM) diodes, MEMS, NEMS

## Abstract

Conventional packaging and interconnection methods for two-terminal devices, e.g., diodes often involve expensive and bulky equipment, introduce parasitic effects and have reliability issues. In this study, we propose a built-in packaging method and evaluate its performance compared to probing and wire bonding methods. The built-in packaging approach offers a larger overlap area, improved contact resistance, and direct connection to testing equipment. The experimental results demonstrate a 12% increase in current, an 11% reduction in resistance, and improved performance of the diode. The proposed method is promising for enhancing sensing applications, wireless power transmission, energy harvesting, and solar rectennas. Overall, the built-in packaging method offers a simpler, cheaper, more compact and more reliable packaging solution, paving the way for more efficient and advanced technologies in these domains.

## 1. Introduction

In the era of the Internet of Things (IoT) and Artificial Intelligence (AI), there has been an increasing demand for devices that are not only smarter but also more efficient. Diodes, essential components in this digital revolution, play a significant role in our increasingly interconnected world. Their asymmetric conductance underpins the Integrated Circuit (IC) industry, enabling the formation of electronic logic. Their ability to allow electrical current to flow predominantly in one direction while obstructing the reverse direction makes them a versatile tool in a diverse range of applications, including power applications such as rectifying antennas (rectennas), high-frequency applications in communication and imaging, and a broad spectrum of sensing technologies.

However, the journey to optimal diodes brings significant challenges. Among these, diode resistance is a crucial factor that impacts detectivity limits and needs to be minimized for superior sensing applications [1]. Power applications, in particular, require efficient rectennas characterized by high responsivity, low resistance, and high asymmetry. Furthermore, low diode resistance is critical for high-frequency applications to maximize efficiency [2]. Balancing these conflicting requirements often presents complex design challenges when crafting ideal diodes [3].

The rising popularity of metal-insulator-metal (MIM) diodes in areas such as energy harvesting [4], solar and infrared rectennas [3,5], wireless power transmission [6], and sensing applications [7,8] is due to their notable advantages. These advantages include a high-frequency response, low power consumption, small size, and cost-effective production [9]. The need for improved current density, resistivity, and signal-to-noise ratio is especially critical in sensing, where enhanced sensitivity, accuracy, and reliability can lead to detecting and measuring weaker signals with greater precision [10]. These advancements in MIM diode characteristics can significantly influence the development of more efficient and innovative techniques.

Building on our recent pioneering work on the built-in packaging of single-terminal devices [11], this study represents a significant advancement in two-terminal device packaging methodologies, explicitly focusing on MIM diodes and directly addressing the above challenges. We introduce an innovative approach to enhance diode performance and reliability that bypasses the limitations associated with traditional packaging. The proposed built-in packaging technique eliminates the need for conventional packaging or a carrier board, thus addressing well-known reliability challenges such as mechanical failure and corrosion associated with wire bonding [12,13,14,15,16,17,18]. By reducing the parasitic resistance arising from conventional packaging and interconnections, we increase current density by 12%, without compromising other essential performance parameters such as asymmetry, nonlinearity, or responsivity.

This novel technique promises significant benefits for high-frequency applications, subdivided into communication and imaging applications [9]. Higher cutoff frequencies facilitate faster data transfer, thereby enhancing communication [19]. Simultaneously, increased detectivity contributes to improved imaging in the terahertz (THz) regime, particularly beneficial for biosensing and astrophysics applications [20,21]. Therefore, our approach to MIM diodes holds the promise for advancing next-generation 6G communication, astrophysics, logic boards, and bioimaging.

In conclusion, our research highlights the potential of integrated packaging in two-terminal devices, presenting opportunities to enhance the efficiency, reliability, and cost-effectiveness of numerous applications. Our successful demonstration of the built-in packaging concept on both single-terminal [11], and two-terminal devices serve as a catalyst for further exploration of its applicability across diverse areas, including sensors, optics, RF, microfluidics, and other related applications.

## 2. Materials and Methods

### 2.1. Fabrication

The self-aligned stepped metal-insulator-metal (MIM) diode configuration was chosen for its precise layer alignment during fabrication, which facilitates efficient electron tunneling through the thin insulator layer. The device area was designed to be 1 μm${}^{2}$ (1 × 1 μm) in order to minimize fabrication costs, which can be achieved by using standard photolithography instead of electron-beam lithography. The Pt-Al electrode combination was selected due to its exceptional work function difference of 1.37 eV, which is one of the highest reported in the existing literature [22], although a difference that exceeds approximately 300 meV may also be considered acceptable [23]. Pt was chosen as the bottom electrode due to its noble metal behavior, exhibiting high oxidation resistance that effectively prevents undesired oxide interference between the bottom and top electrodes, apart from the desired insulator layer. TiO${}_{2}$ was chosen as the insulator layer due to its wide range of applications [24] and promising results in metal-insulator-metal (MIM) configurations with a thickness of 60 nm except for a low current density [2]. To enhance the current density, the thickness of TiO${}_{2}$ was reduced to 6 nm, following a recent study. In particular, a 6 nm thick layer of Al${}_{2}$O${}_{3}$ has demonstrated encouraging results in rectenna applications [22]. The energy diagram, microscope images, and process flow of the fabricated MIM diode are presented in Figure 1 and Table 1, respectively.

A 4-inch prime-grade p-doped silicon wafer with a 50 nm thick thermal oxide layer was selected to serve as the initial substrate. A simple bilayer liftoff technique was employed for the metal electrode patterning process. Initially, the substrate underwent HMDS treatment to enhance adhesion between the substrate and the resist layer. Subsequently, the underlayer (PMGI SF7) was spin coated at a rate of 5000 RPM and subjected to a soft bake at 180 ° C for 5 min. The positive tone UV-resist (Shipley S1805) was then spin-coated at 5000 RPM and soft baked at 120 ° C for 90 s. Using a maskless lithography aligner (MLA) with a laser wavelength of 405 nm and a dose of 80 mJ/cm${}^{2}$, the bottom electrode pattern was exposed. The exposed resist was then developed in MF-319 for one minute and rinsed with deionized water. Subsequently, a 100 nm thick Pt film was deposited onto the wafer at room temperature using the electron beam evaporation technique, with a deposition rate of 0.3 nm/s. To uniformly remove the excess metal, the wafer was immersed in a Remover PG solution overnight, rinsing with acetone and isopropyl alcohol (IPA), and finally dried using a nitrogen gun. The microscope image of the bottom electrode can be seen in Figure 1b.

To create the insulator layer, a 6 nm thick TiO${}_{2}$ film was deposited using the atomic layer deposition (ALD) technique, following the standard recipe described in a recent study [2]. In order to establish electrical contact with the bottom electrode, it was necessary to remove the insulator layer from the contact pads. The process began with HMDS treatment. Later, the positive-tone UV-resist (Shipley S1811) was spin-coated at 5000 RPM and subjected to a soft baking at 120 ° C for 90 s. Using MLA with a laser wavelength of 405 nm and a dose of 100 mJ/cm${}^{2}$, the pattern for the insulator layer was exposed. The exposed resist was then developed in MF-319 for one minute and rinsed with deionized water. The wafer was then immersed in a buffered HF solution to perform a wet etching process, effectively removing excessive TiO${}_{2}$ from the surface.

The top electrode layer was patterned using a single-layer liftoff. The process began by applying HMDS. After that, a positive tone UV resist, Shipley S1805, was spread evenly using spin-coating at a speed of 5000 RPM. The coated resist was then subjected to a soft bake at 120 ${}^{\circ }$C for 90 s. The same MLA parameters as those used for the bottom electrode pattern were applied to define the pattern for the top electrode. Subsequently, the resist exposed to the pattern was developed in MF-319 for one minute, followed by rinsing with deionized water. Then an aluminum film layer with a thickness of 200 nm was deposited onto the wafer at room temperature using electron beam evaporation. The deposition rate was 0.3 nm/s. To remove any excess metal uniformly, the wafer was soaked overnight in a Remover PG solution, then rinsed with acetone and IPA, and finally dried using a nitrogen gun.

Finally, the 4${}^{\prime \prime }$ wafer was diced into smaller dies (8.6 × 8.6 mm) to match the dimensions of a standard SubMiniature version A (SMU) connector jack and a standard chip carrier in order to attach and test the same chip with both. The microscope images and the cross-sectional view of the fabricated MIM diode are shown in Figure 1c. The overlap area of the MIM diode was measured as 1.74 μm${}^{2}$ (1.32 × 1.32 μm).

### 2.2. Packaging and Experimental Setup

Following the fabrication of the chip, IV-curve characterization was carried out using a source measure unit (SMU, B2902A) for each interconnection method as shown in Figure 2.

To evaluate the effectiveness of the proposed built-in packaging method, the same device was characterized using three different techniques: probing with a probe station, wire bonding, and built-in packaging. While soldering is widely used in semiconductor industry, it is not considered suitable for this purpose due to various limitations and potential risks [25,26]. Semiconductor devices are highly sensitive to surface contamination, making them vulnerable to the negative effects of solder flux. Furthermore, the application of thermal stress during soldering can damage delicate components and elevated temperatures may not be compatible with sensitive materials. Diffusion and alloying processes during soldering can also alter the performance of the device. Furthermore, the irreversible nature of soldering limits repair options, and the miniaturization trend poses challenges for reliable solder connections. For these reasons, soldering is not considered.

Probing is a widely utilized technique for preliminary electrical characterization of two- to three-terminal devices, such as diodes or transistors. The experimental setup for probing, as well as the chip used, are shown in Figure 2a. This method possesses two main drawbacks. Firstly, it requires the use of an expensive and bulky setup comprising a probe station, micromanipulators, and a microscope in order to establish electrical contact with the fabricated chip. Second, the probes utilized are thin needle-shaped structures that resemble antennas, which introduce parasitic resistance and capacitance.

For the wire-bonding measurements, the die, Figure 2a, was securely affixed to the chip carrier using double-sided tape. Subsequently, a semi-automatic wedge-wedge bonder (Westbond 4546E) was utilized to wire bond the die using 25 μm Al wire, as shown in the inset of Figure 2b. The chip carrier was then mounted onto a custom PCB, enabling the connection between the chip carrier and the testing equipment. An image of the experimental setup is presented in Figure 2b. While wire-bonding offers a simpler experimental setup compared to probing, the packaging process requires an expensive equipment and a skilled technician to successfully bond the chip to the carrier. Moreover, it is essential to design the die to fit within standardized chip carriers, which imposes restrictions on the dimensions of the die and metal pads. Additionally, it is necessary to design and fabricate the corresponding PCB. Similarly to probing, the wires exhibit antenna-like characteristics that give rise to parasitic effects.

The contact pads of the die were specifically designed to align with and cover the pins of the SMA connector jack. The thickness of the die, 0.5 mm, is slightly smaller than the opening of the SMA connector, therefore the remaining space was fulfilled by placing a foam tape on the backside of the die to provide the necessary support. This approach ensured optimal contact between the connector and the device, allowing reliable connectivity as shown in Figure 2c. The proposed approach has a distinct advantage over probing and wire-bonding methods due to its significantly larger overlap area. This is achieved by establishing a wide contact area between the connector and the device, in contrast to the single point of contact utilized in probing and wire-bonding techniques. The expanded contact area not only improves the reliability and stability of the connection but also reduces the contact resistance. This reduction in contact resistance further contributed to the overall performance and efficiency of the connection. Furthermore, this method offered the advantage of a direct connection between the connector jack and testing equipment, such as an SMU, eliminating the need for additional space or components. Consequently, it was highly space-efficient and required minimal setup.

Finally, the diode was detached from the SMA connector jack and subsequently measured using probing techniques to ensure that its electrical characteristics were unchanged before and after each test procedure.

## 3. Results and Discussion

For the electrical characterization, voltage was swept with the SMU with a step voltage of 4 mV and the current passing through the diode was measured for each interconnection method as described in Figure 2. The measured data were signal processed with the Savitzky-Golay filter in Python to reduce experimental noise and smoothly estimate derivatives of the signal. The plots of the measured IV-curves for each interconnection technique are shown in Figure 3a,b. A uniform increase in the current level of 12% was observed compared to conventional interconnection methods. The maximum current density, denoted as *J*, reached $1.3\times 10^{5}$ using conventional methods. However, the proposed method achieved a higher current density of $1.5\times 10^{5}$, representing an increase of 10%. The resistance of the diode can be found as the slope of the current-voltage curve,
(1)$$R=\frac{dI}{dV}$$
where *I* and *V* are the instantaneous current and voltage, respectively. The calculated values of *R* for each interconnection method are plotted in Figure 3c. The plot clearly shows an average reduction in resistance of 11% compared to conventional interconnection methods. In particular, the zero-bias resistance decreased from 10.7 kΩ to 9.0 kΩ, indicating a substantial decrease of 15%. This is highly significant, since resistance directly impacts various diode performance parameters. First, because of their finite resistance and capacitance *C*, diodes act as a low-pass filter element limiting their operation range by the cut-off frequency, $f_{c}$,
(2)$$f_{c}=\frac{1}{2\pi RC}$$
which suggests that decreasing resistance by 11% corresponds to an increase in the cutoff frequency by 12%. Second, the detectivity of a diode, $D^{*}$, is a measure of its sensitivity to weak signals or its ability to detect low-level signals in the presence of noise. It quantifies the ability of a diode to convert a weak input signal into a detectable electrical output and can be defined as [27]:(3)$$D^{*}=\beta_{\mathrm{sys}}\sqrt{A_{\mathrm{abs}}}\sqrt{R/4kT}$$
where $\beta_{\mathrm{sys}}$ is the system responsivity, $A_{\mathrm{abs}}$ is the absorption area of the detector, *k* is the Bolztmann constant, and *T* is the temperature. As the decreased resistance increases, the signal level, current and detectivity can be increased by 6% by improving the signal-to-noise ratio.

Lastly, for the rectification performances, there are three crucial figures of merit of diodes: asymmetry, nonlinearity, and responsivity. Asymmetry is the ratio between the forward ($I_{F}$) and reverse currents ($I_{R}$):(4)$$\mathrm{Asymmetry}=\;|I_{F}/I_{R}|$$
which needs to be greater than one for the rectification. As shown in Figure 3d, the diode exhibits asymmetric behavior that spans the entire range of 0–1 V, with values exceeding one, demonstrating equivalent responses across the three interconnections. Nonlinearity,
(5)$$\mathrm{Nonlinearity}=\frac{dI}{dV}/\frac{I}{V}$$
indicates the sharpness of the transition from the reverse biased (blocking) state to the forward biased (conducting) state. High nonlinearity is essential to achieve significant rectification. The nonlinearity of the three interconnections shows an equivalent response as plotted in Figure 3e. Responsivity is the ratio of the output DC current to the input AC power and can be quantified with:(6)$$\mathrm{Responsivity}=\frac{d^{2}I}{dV^{2}}/\frac{dI}{2dV}$$
which shows the rectification efficiency of the diode. Figure 3f illustrates the identical responsivity behaviors observed for each method. The average values of the zero-bias responsivity were found to be 0.55 V${}^{-1}$, while the maximum responsivity reached 14.3 V${}^{-1}$ at 1 V.

The experimental data are provided in the Appendix A. Additionally, the Python code utilized for signal and data processing can be found in the Appendix B. In order to assess the performance of the fabricated diode, a comparison is made in Table 2 between its performance parameters and those of the most advanced MIM diodes reported in recent literature.

The performance of the fabricated and tested diode is similar to those reported in the literature. This suggests that similar improvements can be achieved for state-of-the-art diodes without sacrificing any performance parameters by implementing the two-terminal built-in packaging approach.

In sensing applications, achieving an improved signal-to-noise ratio is essential to enhance detectivity and sensitivity. This improvement enables more reliable detection and measurement of weaker signals [34]. Likewise, in wireless power transmission, minimizing power loss during energy transfer is crucial, and this is achieved by reducing resistance. This, in turn, enhances the efficiency of wireless charging systems [35].

In addition, in energy harvesting, increasing the current density plays a significant role in reducing energy loss and improving the efficient conversion of ambient energy sources into usable electrical power [4,9,36].

Furthermore, the proposed built-in packaging approach offers several advantages in the test setup. These include simplicity, cost-effectiveness, compact size, and robustness. These advancements open up new possibilities for the development of more efficient technologies in energy harvesting, solar rectennas, wireless power transmission, and sensing applications.

## 4. Conclusions

Our study introduces an innovative approach to diode packaging, designed to improve performance and reliability by overcoming traditional limitations. Our proposed built-in packaging method negates the need for conventional packaging or carrier boards, which are known for their susceptibility to mechanical failure and corrosion. The experimental results reveal the potential of our built-in packaging method. Improvements in current levels, reduction in resistance, and enhanced performance parameters promise substantial advantages for a multitude of applications, notably in sensing, wireless power transmission, energy harvesting, and solar rectennas.

With the rapid advancements in technological sectors, the contribution of our study becomes all the more significant. The built-in packaging method invites simpler, cheaper, and more reliable packaging solutions. This method not only proposes a more efficient approach to existing challenges, but also paves the way for future technological advancements in these domains. However, it is important to note that reliability tests have not been conducted in this work, highlighting the need for further exploration of the reliability of this structure.

In conclusion, our research provides significant progress in the field of two-terminal device packaging, presenting built-in packaging as a viable solution to enhance efficiency, reliability, and cost-effectiveness in numerous applications. It marks a significant contribution to the broader advancements in diode design and packaging techniques, using an innovative approach. The success of the built-in packaging concept in both single-terminal and two-terminal devices paves the way for further exploration of its applicability across a diverse range of areas, including sensors, optics, RF, microfluidics, and more. Thus, the implications of our research extend beyond the current study, stimulating thought and potential future research into improving the efficiency of two-terminal devices and their applications.   

## Figures and Tables

**Figure 1 micromachines-14-01473-f001:**
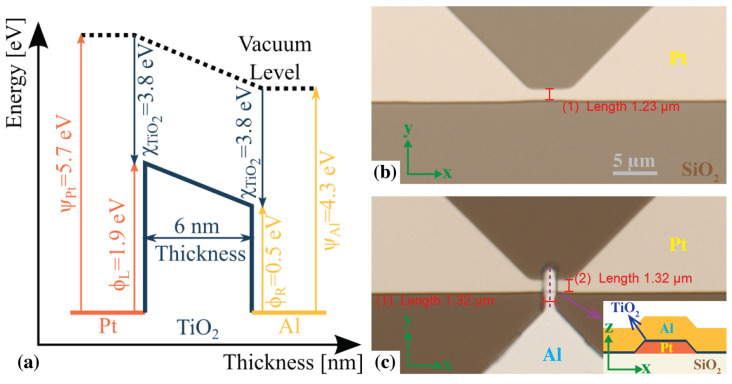
(**a**) Energy diagram for the Pt-TiO${}_{2}$-Al MIM diode, and microscope images of the device before (**b**) and after (**c**) the top electrode deposition (the inset shows the cross-sectional view).

**Figure 2 micromachines-14-01473-f002:**
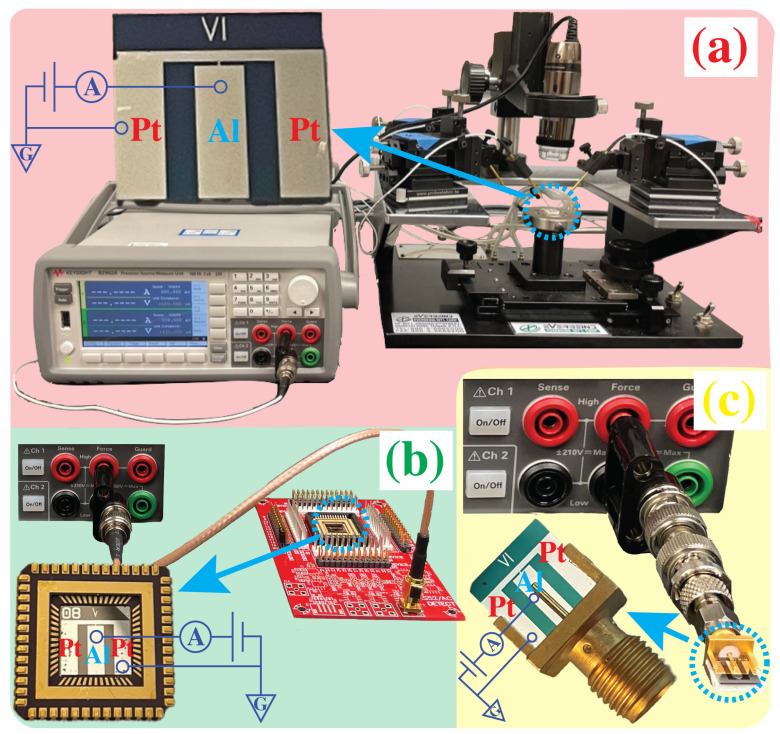
Experimental setup and electrical schematics of (**a**) probing, (**b**) wire-bonding, and (**c**) built-in packaging.

**Figure 3 micromachines-14-01473-f003:**
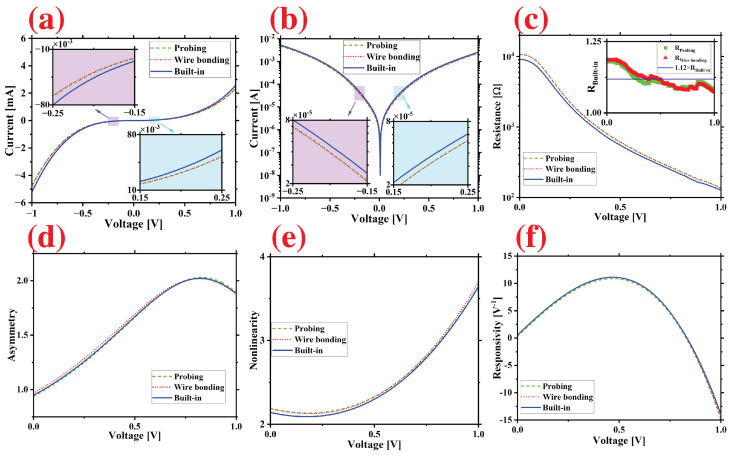
Electrical characteristics of the fabricated diode using three different interconnection methods: (**a**) linear and (**b**) logarithmic scale IV-curves. The diode resistance is shown on logarithmic scale in (**c**). The inset depicts the ratio of resistance between conventional methods and built-in packaging. The diode’s asymmetry, nonlinearity, and responsivity are shown in (**d**–**f**), respectively.

**Table 1 micromachines-14-01473-t001:** Fabrication steps for MIM diode.

Layer	Step 1	Step 2	Step 3	Step 4
Bottom electrode	PR coating	Lithography	Deposition	Litfoff
Insulator	Deposition	PR coating	Lithography	Wet etch
Top electrode	PR coating	Lithography	Deposition	Litfoff

**Table 2 micromachines-14-01473-t002:** Performance comparison of the MIM diodes with the literature.

Structure	Insulator Thickness [nm]	Area [μm${}^{2}$]	Asym. ≈0.5 V	Max. Nonlin.	Zero-Bias Respon [V${}^{-1}$]	Max. Respon. [V${}^{-1}$]	Max. *J* [A/cm${}^{2}$]	Zero-Bias Resistance [Ω]	Application	Interconnection	Ref.
Al/AlO${}_{x}$/Ag	0.75	$1.76\times 10^{6}$	1.2	12.5	9	9	$3\times 10^{-10}$	$27\times 10^{3}$	Rectenna	Probing	[28]
Al/AlO${}_{x}$/Pt	0.1–0.2	$8\times 10^{-3}$	1	$9.9\times 10^{-6}$	-	$10^{-3}$	$43\times 10^{6}$	$1.25\times 10^{2}$	IR-detector	Probing	[29]
Al/AlO${}_{x}$/Pt	1–2.5	$4\times 10^{-3}$	-	-	-	4.8	$5\times 10^{2}$	$312\times 10^{6}$	IR-detector	Probing/Wire bonding	[30]
Al/AlO${}_{x}$/Pt	2.5–3.5	$4\times 10^{-3}$	1	$3\times 10^{-3}$	-	$10^{-2}$	$500\times 10^{3}$	$1.65\times 10^{3}$	IR-detector	Probing/Wire bonding	[31]
Nb/NbO${}_{x}$/Pt	15	$6.4\times 10^{3}$	$7.7\times 10^{3}$	4.7	15	16.9	$1\times 10^{-2}$	$7.8\times 10^{9}$	-	Probing	[32]
Ni/NiO/Pt	1–2	$2.5\times 10^{-3}$	-	9	-	6.5	$3.6\times 10^{-2}$	≈$10^{7}$	IR-detector	Probing	[33]
Al/TiO${}_{2}$/Pt	6	1.74	1.7	3.7	0.55	14.3	$1.3\times 10^{5}$	$10.7\times 10^{3}$	-	Probing	ours
Al/TiO${}_{2}$/Pt	6	1.74	1.7	3.7	0.55	14.3	$1.3\times 10^{5}$	$10.7\times 10^{3}$	-	Wire bonding	ours
Al/TiO${}_{2}$/Pt	6	1.74	1.7	3.7	0.55	14.3	$1.5\times 10^{5}$	$9.0\times 10^{3}$	-	Built-in	ours

## Data Availability

The experimental data used in this work is provided in the Appendix A.

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
