# Peer review of "Built-In Packaging for Two-Terminal Devices"

_micromachines, 2023, doi:10.3390/mi14071473_

Round 1
Reviewer 1 Report
The author has presented a quick built-in packaging that can install MIM sensors with only an insertion of properly designed pad structures. In this new work, two-terminal structures with different metal have been fabricated and validated that it can extend from its prior research. The process is easy, cost effective, and providing more resistance to the environment for the sensors. The reviewer believes the work is well-prepared and experiment result is backed up with sufficiently explanations that justifies for the publication in micromachines.
The reviewer has some minor comments on the content of the work, of which the reviewer believe will help the broad audience in the community understand the work more if addressed.
1. Table 2 should summarizes or categorize the literature based on its feature or application. A grouping and separation between each application would be good for detail compare of this work and others.
2 Al and Pt has very different optical reflection on oxide at their thickness of 200nm/100nm the image of pad the device looks exactly same in color in Fig. 2. Was it dummy sample?
3. While soldering can resolve most of the issue, built-in packaging provides accessibility and replace-capability, The author has considered two types of the packaging approaches already. Considering the technology has been used from DC to several hundreds GHz. An explanation to not to consider it or compare with soldering would be good.
4. TiO2 is pretty leaky and the design of using only 6nm for the MIM structures should be explained or referenced by literature.
5. No reliability test of this structure in this work or in prior work(MDPI 2022) has been presented. The claim in the conclusion should be considered re-wording.
Author Response
Thanks for your great feedback!
- Table 2 has been updated, and two columns were added as applications and interconnections,
- The microscope images were taken from the real sample. The bottom electrode, Pt, was coated with 6-nm thick TiO2, while the top electrode, Al, wasn't. Maybe the thin TiO2 layer decreases the contrasts between the two different metals. The image was taken using Olympus MX-61A. The exposure time and intensity of the light could play some role to get such a low contrast.
- The following section is added to page #5 line #129,
- ...While soldering is widely used in semiconductor industry, it is not considered suitable for this purpose due to various limitations and potential risks [25,26]. Semiconductor devices are highly sensitive to surface contamination, making them vulnerable to the negative effects of solder flux. Furthermore, the application of thermal stress during soldering can damage delicate components and elevated temperatures may not be compatible with sensitive materials. Diffusion and alloying processes during soldering can also alter the performance of the device. Furthermore, the irreversible nature of soldering limits repair options, and the miniaturization trend poses challenges for reliable solder connections. For these reasons, soldering is not considered....
- The following section was added to page #2, line #75,
- ...TiO2 was chosen as the insulator layer due to its wide range of applications [24] and promising results in MIM configurations with a thickness of 60 nm except for a low current density [2]. To enhance the current density, the thickness of TiO2 was reduced to 6 nm, following a recent study. In particular, a 6 nm thick layer of Al2O3 has demonstrated encouraging results in rectenna applications [22]....
- The following section was added to page #8, line #253,
- ...However, it is important to note that reliability tests have not been conducted in this work, highlighting the need for further exploration of the reliability of this structure....

Reviewer 2 Report
The study was organized well for publication. I have no comment on this study.
Author Response
Thank you!